# FLASHSAMPLING: FAST AND MEMORY-EFFICIENT EXACT SAMPLING WITH GROUP-GUMBEL-MAX

## ABSTRACT

Sampling operations in discrete space are widely used in different fields such as language models, reinforcement learning, VAE, GAN, and neural architecture search. Current sampling methods involve computing the softmax operation across the entire categories, leading to significant computational and memory requirements, particularly when dealing with large sampling categories. This paper presents a novel sampling approach known as FlashSampling, designed to alleviate the computational and communication overhead by circumventing the computation of the softmax operation. Our method maintains mathematical equivalence to conventional sampling strategies while demonstrating significantly enhanced speed and memory efficiency. This is achieved by partitioning the category into distinct groups for independent sampling and then leveraging the Gumble-Max trick to eliminate the need for softmax computation. We substantiate the correctness and efficacy of our method both through mathematical proofs and empirical validation. Extensive experimental outcomes illustrate marked enhancements in speed and memory utilization, with FlashSampling attaining up to 384% faster sampling times and 1822% reduced memory consumption.

## 1 INTRODUCTION

Sampling in discrete spaces is fundamental to a wide array of machine learning domains, including language modeling (Brown et al., 2020), reinforcement learning (Mnih et al., 2013), variational autoencoders (VAE) (van den Oord et al., 2017), generative adversarial networks (GAN) (Yu et al., 2016), and neural architecture search (Zoph & Le, 2017). In language models, discrete sampling is indispensable for generating coherent and contextually relevant text by selecting words from a vast vocabulary Sutskever (2014). Reinforcement learning algorithms rely on sampling actions from policy distributions to explore and learn optimal strategies within complex environments Mnih et al. (2015). VAEs and GANs employ sampling techniques to generate new data instances from learned latent spaces, facilitating tasks like image synthesis and data augmentation van den Oord et al. (2017); Yu et al. (2016). Neural architecture search utilizes sampling to efficiently explore a vast space of possible network architectures, aiming to discover models with superior performance while minimizing computational resources (Zoph & Le, 2017).

Typically, sampling from a categorical distribution involves computing the Softmax function to obtain probabilities over all possible categories. As the number of categories increases, this approach becomes computationally intensive and memory-demanding due to the need to calculate the Softmax denominator and store the full set of probabilities for multinomial sampling. The complexity poses challenges, particularly in auto-regressive architectures where each token is sequentially generated based on previously produced tokens (Brown et al., 2020). While numerous acceleration algorithms have been developed for continuous space sampling in diffusion models (Neal, 2012; Hoffman et al., 2014; Tucker et al., 2017; Grathwohl et al., 2017; CORNUET et al., 2012), discrete space sampling remains relatively under-explored (Jang et al., 2016; Kool et al., 2019).

The mainstream method still involves first using Softmax to compute probabilities and then performing sampling. These challenges highlight the necessity for more efficient sampling techniques that can bypass the computational bottlenecks of the Softmax operation, reduce storage overhead, and maintain mathematical correctness and performance. This raises the question: *can we address*

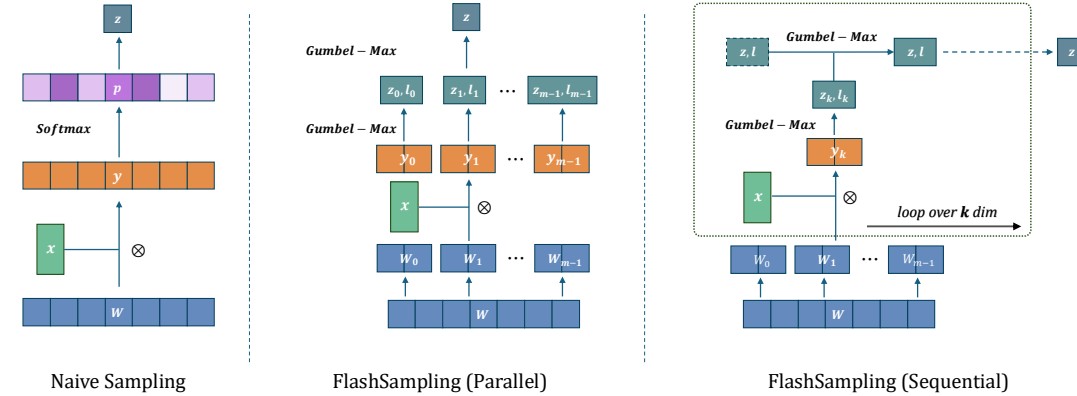

Figure 1: **Operational illustration of FlashSampling**. From left to right: naive sampling, Flash-Sampling(parallel), and FlashSampling(sequential) using online computing. $\mathbf{x} \in \mathbb{R}^d$ denotes embedding, $\mathbf{W} \in \mathbb{R}^{d \times V}$ denotes category projection matrix, $\mathbf{y}$ denotes logits, $\mathbf{p}$ denotes probability, $z$ denotes the sampling result and $l$ denotes the intermediate variables in the FlashSampling process.

*both computational and memory issues simultaneously while performing accurate multinomial sampling?*

In this paper, we introduce the FlashSampling algorithm, an exact sampling method to sample categorical distribution that simultaneously addresses both computational efficiency and memory overhead issues. To tackle the first issue—the significant computational burden of calculating the Softmax denominator—we employ the Gumbel-Max trick (Jang et al., 2016) This technique allows us to sample from categorical distribution using only the logits (the values before applying Softmax), eliminating the need to compute the Softmax function. To resolve the second issue of substantial memory requirements, we implement a two-stage group sampling strategy. In the first stage, we conduct intra-group sampling within each group; in stage two, we perform inter-group sampling on the candidates selected from each group. This approach reduces the storage complexity from $O(V)$ to $O(V/g)$ in the parallel version or even to $O(g)$ in the sequential version, where $V$ is the category size and $g$ is the number of groups. FlashSampling can be easily extended to a distributed version, where the communication overhead is reduced to $O(1)$. This significant reduction in communication cost, independent of the category size, makes it highly efficient for distributed settings.

We validated the effectiveness of FlashSampling across various scenarios, including speed and memory tests as well as generation quality tests. Specifically, we conducted standalone tests (focusing solely on the sampling function) and end-to-end tests (LLM inference) to compare FlashSampling with the baseline in terms of speed and memory consumption. Additionally, in the end-to-end tests, we evaluated the generation quality of FlashSampling against the baseline. FlashSampling demonstrated faster performance, lower memory consumption, and comparable generation quality to the baseline.

## 2 RELATED WORK

### 2.1 SAMPLING IN DEEP-LEARNING

**Sampling in Contiguous Space.** Various statistical methods have been developed to improve sampling from continuous distributions. Building upon the foundation of Markov Chain Monte Carlo (MCMC) (Gamerman & Lopes, 2006), advanced techniques such as Hamiltonian Monte Carlo (HMC) Neal (2012) and the No-U-Turn Sampler (NUTS) Hoffman et al. (2014) enhance convergence and efficiency by leveraging gradient information and adaptively adjusting path lengths during sampling. For large-scale datasets, methods like Stochastic Gradient Langevin Dynamics (SGLD) Welling & Teh (2011) and Stochastic Gradient Hamiltonian Monte Carlo (SGHMC) Chen et al. (2014) reduce computational overhead by incorporating stochastic gradients and mini-batch data while still capturing model uncertainty.

**Sampling in Discrete Space.** Optimizing sampling for discrete variables and categorical distributions presents significant challenges due to the absence of gradient information. To tackle these

challenges, methods such as Sequential Monte Carlo (SMC) Doucet (2001), Particle Gibbs sampling Andrieu et al. (2010), REBAR Tucker et al. (2017), and RELAX Grathwohl et al. (2017) have been developed. These methods aim to improve computational efficiency, reduce variance, and enable gradient-based optimization in models with discrete variables. Additionally, Adaptive Importance Sampling CORNUET et al. (2012) and enhanced categorical rejection sampling methods Neumann (1951); Efraimidis (2015) increase efficiency by adjusting proposal distributions and optimizing acceptance probabilities, particularly when dealing with high-dimensional categorical data. The Gumbel-Max trick Jang et al. (2016) is another effective technique for sampling from categorical distributions. It transforms the sampling process into a maximization problem by adding Gumbel noise to the log probabilities of the categories and selecting the category with the maximum perturbed value. This approach is particularly useful for discrete latent variable models in variational inference. Parallelization strategies such as Parallel Tempering and Replica Exchange MCMC Earl & Deem (2005) enhance exploration efficiency by running multiple chains at different temperatures in parallel. Furthermore, Variational Inference (VI) Blei et al. (2017) transforms the sampling problem into an optimization task, providing faster convergence with some bias. Collectively, these methods significantly enhance the practicality and scalability of sampling in deep learning.

## 2.2 MEMORY-EFFICIENT METHOD

Memory-efficient methods are extensively employed in attention computation, which is computationally intensive and involves significant memory I/O operations. Online softmax approach (Rabe & Staats, 2021) is introduced to efficiently compute numerically stable attention scores sequentially while maintaining linear memory usage. To address time and memory consumption during training, FlashAttention (Dao et al., 2022; Dao, 2023) utilizes tiling strategies to minimize memory reads and writes between the GPU's high-bandwidth memory (HBM) and on-chip SRAM. xFormers also introduces a similar technique (Lefaudeux et al., 2022). PagedAttention (Kwon et al., 2023) optimizes the use of the KV cache memory by reducing waste and enabling adaptive sharing among batched requests during inference. Furthermore, similar grouping and tiling approaches are used by techniques such as Lightning Attention (Qin et al., 2024) and Flash Linear Attention (Yang et al., 2023; 2024; Zhang et al., 2024) to optimize GPU memory consumption in linear-complexity attention mechanisms.

## 3 METHOD

---

**Algorithm 1** FlashSampling(Parallel)

**Input:** $\mathbf{x} \in \mathbb{R}^d$, $\mathbf{W} \in \mathbb{R}^{d \times V}$, group size $g$, flag.

Divide $\mathbf{W}$ into $m = n/g$ blocks $\mathbf{W}_0, \mathbf{W}_1, ... \mathbf{W}_{m-1}$ of size $d \times g$ each.

Compute $\mathbf{y}_k = \mathbf{W}_k^\top \mathbf{x} \in \mathbb{R}^g, k = 0, \ldots, m-1$.

Sample
$$z_k = \arg\max_j y_{kj} - \log(-\log u_{kj}),$$
$$u_{kj} \overset{i.i.d.}{\sim} U(0,1).$$

Sample
$$i = \arg\max_k l_k - \log(-\log \bar{u}_k),$$
$$\bar{u}_k \overset{i.i.d.}{\sim} U(0,1).$$

**if** flag **then**
  $l = \text{lse}([l_0, \ldots, l_{m-1}])$.
**else**
  $l = -\infty$.
**end if**
Return $z = z_i, l$.

---

**Algorithm 2** Flash Sampling(Sequential)

**Input:** $\mathbf{x} \in \mathbb{R}^d, \mathbf{W} \in \mathbb{R}^{d \times V}$, group size $g$,flag.

Divide $\mathbf{W}$ into $m = \frac{n}{g}$ blocks $\mathbf{W}_0, \mathbf{W}_1, ... \mathbf{W}_{m-1}$ of size $d \times g$ each.

Initialize $l = -\infty, z = 0$.

**for** $k = 0, \ldots, m-1$ **do**
  $\mathbf{y}_k = \mathbf{W}_k^\top \mathbf{x} \in \mathbb{R}^g$.
  Sample
  $$z_k = \arg\max_j y_{kj} - \log(-\log u_{kj}),$$
  $$u_{kj} \overset{i.i.d.}{\sim} U(0,1).$$
  $l_k = \text{lse}(\mathbf{y}_k)$.
  $\bar{l} = [l, l_k]$.
  Sample
  $$i_k = \arg\max_j \bar{l}_j - \log(-\log \bar{u}_j),$$
  $$\bar{u}_j \overset{i.i.d.}{\sim} U(0,1).$$
  $z = [z, z_k]_{i_k}$.
  $l = \text{lse}(\bar{l})$.
**end for**
**if** not flag **then**
  $l = -\infty$.
**end if**
Return $z, l$.

---

In this section, we'll explore sampling categorical distribution in deep learning and introduce our proposed method FlashSampling using the Group-Gumbel-Max trick. We'll examine its parallel, sequential, and distributed implementations.

In the following discussion, we assume $d$ represents the number of features, $V$ represents the number of categories, $\mathbf{x} \in \mathbb{R}^d$ denotes a column vector, and $\mathbf{W}$ denotes a matrix. We use $C(\mathbf{p})$ to represent a Categorical distribution, where $\mathbf{p} \in \mathbb{R}^V$ and $\sum_{j=0}^{V-1} p_j = 1$. We use lse to represent the 'LogSumExp' function, defined as $\mathrm{lse}(\mathbf{x}) = \log\left(\sum_j \exp(x_j)\right)$.

## 3.1 SAMPLING CATEGORICAL DISTRIBUTIONS IN DEEP LEARNING

In deep learning, sampling categorical distributions is typically performed through the following steps: first, we obtain features $\mathbf{x} \in \mathbb{R}^d$ and a categorical projection $\mathbf{W}^{d \times V}$ from the neural network. Using matrix multiplication, we compute the logits $\mathbf{y} = \mathbf{W}^\top \mathbf{x} \in \mathbb{R}^V$. Then, the 'Softmax' function is applied to compute the probability distribution $\mathbf{p} = \mathrm{Softmax}(\mathbf{y})$, and finally, sampling is performed.

---

**Algorithm 3** FlashSampling(Distrubuted)

---
**Input:** $\mathbf{x} \in \mathbb{R}^d, \mathbf{W} \in \mathbb{R}^{d \times V}$, distributed world size $n$, group size $g$.
Divide $\mathbf{W}$ into $n$ blocks $\mathbf{W}_0, \mathbf{W}_1, ... \mathbf{W}_{n-1}$ of size $d \times (V/n)$ each.
$z_k, l_k = \mathrm{flash\_sampling}(\mathbf{x}, \mathbf{W}_k, g, \mathrm{True})$.
$\mathbf{z}, \mathbf{l} = \mathbf{0} \in \mathbb{R}^n$.
$\mathrm{Gather}(z_k, \mathbf{z}, \mathrm{dst} = 0)$.
$\mathrm{Gather}(l_k, \mathbf{l}, \mathrm{dst} = 0)$.
On rank0, sample $i = \arg\max_k l_k - \log(-\log u_k)$, $k = 0, \ldots, n-1$.
Return $z_i$.

---

From the above process, it is clear that in deep learning, before sampling, we need to compute the probability distribution using 'Softmax' and store it. This differs from typical sampling categorical distributions scenarios, where we are directly given the probability distribution $\mathbf{p} = (p_0, \ldots, p_{V-1})$, and there is no need to compute $\mathbf{p}$.

---

**Algorithm 4** Gumbel-Max sampling

---
**Input:** $\mathbf{x} \in \mathbb{R}^d, \mathbf{W} \in \mathbb{R}^{d \times V}$.
Compute $\mathbf{y} = \mathbf{W}^\top \mathbf{x} \in \mathbb{R}^V$.
Compute $z_k = y_k - \log(-\log u_k), u_k \stackrel{i.i.d.}{\sim} U(0,1), k = 0, \ldots, V-1$.
Return $z = \arg\max_k z_k$.

---

This approach poses two main issues when the category size is large:

- Computing the probability requires computing the Softmax function, which introduces significant computational overhead.

- The need to store the probability distribution $\mathbf{p}$ to perform sampling, leads to a high memory demand.

These challenges lead us to the following question: Can we address both of these issues and still perform accurate sampling from Categorical distribution? In the following subsections, we will discuss how to solve these two problems.

## 3.2 FLASHSAMPLING WITH GROUP-GUMBEL-MAX

### 3.2.1 USING GUMBEL-MAX TO AVOID COMPUTING SOFTMAX

Our first optimization is to use the Gumbel-Max (Jang et al., 2016) trick to avoid computing Softmax. Sampling $z \sim C(\mathbf{p})$ is equivalent to:

$$z = \arg\max_k \left[\log p_k - \log(-\log u_k)\right], \quad u_k \stackrel{i.i.d.}{\sim} U(0,1) \tag{1}$$

where $U(0,1)$ represents the uniform distribution over $(0,1)$, and $i.i.d.$ stands for independent and identically distributed.

Substituting $p_k = \frac{\exp(y_k)}{\sum_j \exp(y_j)}$ into the equation, we get:

$$z = \arg\max_k \left[\log p_k - \log(-\log u_k)\right]$$

$$= \arg\max_k \left[\log\exp(y_k) - \log\left(\sum_j \exp(y_j)\right) - \log(-\log u_k)\right] \quad (2)$$

$$= \arg\max_k \left[y_k - \text{lse}(\mathbf{y}) - \log(-\log u_k)\right]$$

$$= \arg\max_k \left[y_k - \log(-\log u_k)\right], \quad u_k \overset{i.i.d.}{\sim} U(0,1).$$

We summarize this in the following proposition:

**Proposition 3.1.** *Sampling $z \sim C(\mathbf{p})$ is equivalent to:*

$$z = \arg\max_k \left[y_k - \log(-\log u_k)\right], \quad u_k \overset{i.i.d.}{\sim} U(0,1).$$

By using this proposition, we can perform sampling without needing to compute the Softmax function, although we still need to calculate the complete logits.

### 3.2.2 USE GROUP TECHNIQUE TO AVOID MATERIALIZING LOGITS

The second optimization is based on the following fact:

$$p_k = \frac{\exp(x_k)}{\sum_j \exp(x_j)} = \frac{\exp(x_k)}{\sum_{j \in A} \exp(x_j)} \cdot \frac{\sum_{j \in A} \exp(x_j)}{\sum_j \exp(x_j)}.$$

where $A$ is any subset that contains $k$. The intuitive meaning of this equation is that sampling from category distribution can be decomposed into two steps: first, sampling a subset $A$, and then sampling within subset $A$. Based on this fact, we present the following proposition:

**Proposition 3.1.1.** *Given a Categorical distribution $C(\mathbf{p})$ and group size $g$, sampling from $C(\mathbf{p})$ is equivalent to sampling $z_k$ from $C(\mathbf{p}_k), k = 1, \ldots, V/g - 1$, sampling index $i$ from $C(\bar{\mathbf{p}})$, and give $z_i$ as the result. Where*

$$p_{kj} = e_{kj}/e_k, \bar{p}_k = e_k/e,$$

$$e_{kj} = \exp(y_{A_{kj}}), e_k = \sum_{j \in A_k} \exp(y_j), e = \sum_k e_j, p_{kj} = e_{kj}/e,$$

$$A_k = \{kg, \ldots, (k+1)g - 1\}, A_{kj} = kg + j, k = 0, \ldots, V/g - 1, j = 0, \ldots, g - 1.$$

If the process of sampling $z_k$ from $C(\mathbf{p}_k)$ is done in parallel, we obtain a parallel algorithm 5. If done sequentially, we obtain a sequential algorithm 6. The proof of correctness for the sequential version is as follows:

*Proof of Algorithm 6.*

$$\mathbf{P}(z = A_{kj}) = \mathbf{P}(z_k = A_{kj}) \cdot \mathbf{P}(i_k = 1) \prod_{s=k+1}^{V/g-1} \mathbf{P}(i_s = 0)$$

$$= \frac{e_{kj}}{e_k} \cdot \frac{e_k}{\sum_{s \leq k} e_s} \prod_{s=k+1}^{V/g-1} \frac{\sum_{t \leq s-1} e_t}{\sum_{t \leq s} e_t} = \frac{e_{kj}}{e_k} \cdot \frac{e_k}{e} = \frac{e_{kj}}{e}. \quad (3)$$

$\square$

Algorithm 5, 6 allows for efficient sampling without materializing the complete distribution, reducing memory requirements.

| **Algorithm 5** Group Softmax Sampling (Par.) | **Algorithm 6** Group Softmax Sampling (Seq.) |
|---|---|
| **Input:** $\mathbf{x} \in \mathbb{R}^d, \mathbf{W} \in \mathbb{R}^{d \times V}$, group size $g$. | **Input:** $\mathbf{x} \in \mathbb{R}^d, \mathbf{W} \in \mathbb{R}^{d \times V}$, group size $g$. |
| Divide $\mathbf{W}$ into $m = \frac{n}{g}$ blocks $\mathbf{W}_0, \mathbf{W}_1, \dots \mathbf{W}_{m-1}$ of size $d \times g$ each. | Divide $\mathbf{W}$ into $m = \frac{n}{g}$ blocks $\mathbf{W}_0, \mathbf{W}_1, \dots \mathbf{W}_{m-1}$ of size $d \times g$ each. |
| Compute $\mathbf{y}_k = \mathbf{W}_k^\top \mathbf{x} \in \mathbb{R}^g, k = 0, \dots, m-1$. | Initialize $l = -\infty, z = 0$. |
| Compute $e_{kj} = \exp(y_{kj}), e_k = \sum_j e_{kj}, e = \sum_k e_k$. | **for** $k = 0, \dots, m-1$ **do** |
| Compute $p_{kj} = e_{kj}/e_k, \bar{p}_k = e_k/e, \mathbf{p}_k = (p_{k0}, \dots, p_{k,g-1}), \bar{\mathbf{p}} = (\bar{p}_0, \dots, \bar{p}_{m-1}), k = 0, \dots, m-1, j = 0, \dots, g-1$. | $\quad \mathbf{y}_k = \mathbf{W}_k^\top \mathbf{x} \in \mathbb{R}^g$. |
| | $\quad l_k = \text{lse}(\mathbf{y}_k)$. |
| | $\quad \mathbf{p}_k = \exp(\mathbf{y}_k)/\exp(l_k)$. |
| | $\quad$ Sample $z_k \sim C(\mathbf{p})$. |
| | $\quad l'_k = \text{lse}([l, l_k])$. |
| | $\quad \bar{\mathbf{p}}_k = [\exp(l)/\exp(l'_k), \exp(l_k)/\exp(l'_k)]$. |
| Sample $z_k \sim C(\mathbf{p}_k), k = 0, \dots, m-1$. | $\quad$ Sample $i_k \sim C(\bar{\mathbf{p}}_k)$. |
| Sample $i \sim C(\bar{\mathbf{p}})$. | $\quad z = [z, z_k]_{i_k}$. |
| Return $z = z_i$. | $\quad l = l'_k$. |
| | **end for** |
| | Return $z$. |

### 3.3 Put every thing together

It is important to note that in the previous section, both $\mathbf{p}_k$ and $\bar{\mathbf{p}}$ are categorical distributions, allowing us to apply the Gumbel-Max trick. By combining the two previously mentioned tricks, we can derive FlashSampling(parallel) and FlashSampling(sequential). The complete algorithms are detailed in Algorithm 1, 2.

### 3.4 Extension: Distributed Version

Scalability is becoming increasingly important, and FlashSampling naturally extends to a distributed version.

Let's first describe the traditional distributed sampling method: Suppose the number of categories is $V$, $\mathbf{W}$ is the category projection matrix, $\mathbf{x}$ is the embedding, and the distributed world size is $n$. In traditional distributed sampling, GPU with rank $k$ stores a slice of the weight matrix $\mathbf{W}_k = \mathbf{W}[:, k \times V/n : (k+1) \times V/n], k = 0, \dots, n-1$. Each GPU with rank $k$ computes its logits $y_k$ independently, followed by a 'gather' operation where the logits are gathered to rank 0. Rank 0 then performs the categorical sampling. The PyTorch-like code for this process is as follows:

```python
def dist_sample(x, W):
    y = F.linear(x, W)
    y_array = [torch.empty_like(y) for _ in range(world_size)]
    dist.gather(y, gather_list=y_array)
    y = torch.cat(y_array, dim=-1)
    prob = F.softmax(y, dim=-1)

    return torch.multinomial(prob, num_samples=1)
```

As seen in the code, due to the need to communicate logits across GPUs, the communication complexity is $O(V)$, which results in significant communication overhead.

By leveraging the Group Technique, we can first perform local sampling on each GPU and then communicate only the sampled results. Then, a sampling step can be performed to get the final result. The PyTorch-like code for this approach is as follows:

```python
def dist_sample(x, W):
    id, l = flash_sampling(x, W)
    id_array = [torch.empty_like(id) for _ in range(world_size)]
    l_array = [torch.empty_like(l) for _ in range(world_size)]
    dist.gather(id, gather_list=id_array)
    dist.gather(l, gather_list=l_array)
    id = torch.cat(id_array, dim=-1)
    l = torch.cat(l, dim=-1)
    output = gumbel_max_sampling(id, l)

    return output
```

Figure 2: **Time Comparison (measured in ms) of Naive and Parallel FlashSampling Across Sequence Lengths and Hidden Dimensions for Batch Size 2048**. Each sub-figure represents the performance across hidden dimensions of 256, 512, 1024, and 2048. Dashed lines highlight where FlashSampling significantly surpasses Naive Sampling in time efficiency. The smaller the metric, the better.

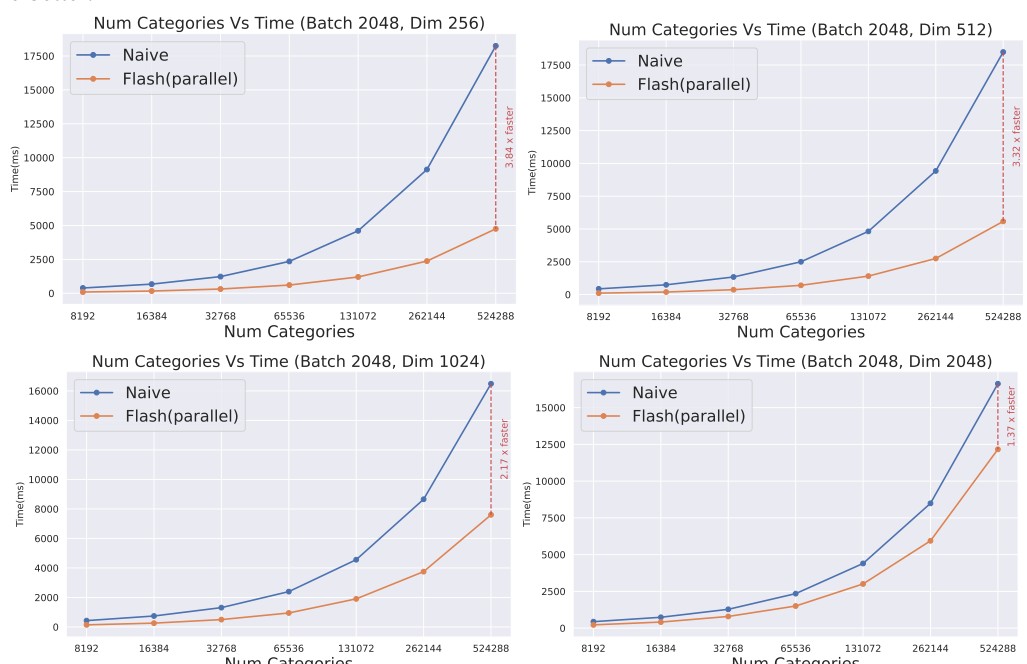

Here, $l$ represents the local logit at each GPU, and the complete algorithm is outlined in Algorithm 3. Notice that with this method, the communication complexity is reduced to $O(1)$, greatly improving communication efficiency.

Table 1: **Time Comparison of Naive Sampling, FlashSampling(Parallel), and FlashSampling(Sequential) across category Sizes (8K-512K) in 128 and 256 Dimensions**. Dashes indicate out-of-memory errors, and times in the table are measured in seconds (**s**). The smaller the metric, the better.

| Method\Vocab. | Dim | 8,192 | 16,384 | 32,768 | 65,536 | 131,072 | 262,144 | 524,288 |
|---|---|---|---|---|---|---|---|---|
| Naive | 128 | 8.85 | 17.19 | 34.85 | 71.03 | 142.76 | - | - |
| Flash-Parallel | 128 | 2.50 | 4.94 | 9.94 | 19.67 | 39.67 | 78.68 | 157.61 |
| Flash-Sequential | 128 | 9.20 | 18.40 | 36.71 | 72.62 | 144.88 | 291.25 | 580.35 |
| Naive | 256 | 10.18 | 19.35 | 39.18 | 76.34 | 150.67 | - | - |
| Flash-Parallel | 256 | 3.13 | 6.23 | 12.70 | 25.20 | 50.89 | 100.53 | 202.73 |
| Flash-Sequential | 256 | 10.17 | 20.31 | 40.73 | 81.50 | 163.27 | 326.06 | 647.98 |

## 4 EXPERIMENTS

### 4.1 FAST SAMPLING: STANDALONE COMPARISON

In this section, we present a detailed standalone test comparison among FlashSampling(parallel), FlashSampling(sequential), and Naive Sampling. We evaluate and contrast their performance by examining variations in speed and memory consumption across different numbers of category sizes and hidden dimensions. The detailed comparisons are vividly depicted in Fig. 2 and Fig. 3, with extensive details provided in Table 1 and Table 2.

From the analysis of time and memory consumption illustrated in Fig. 2 and Fig. 3, it is evident that FlashSampling(parallel) achieves a speed up to 3.8 times faster than Naive Sampling and uses

Figure 3: **Memory Consumption (measured in GB) of Naive and Parallel FlashSampling Across Sequence Lengths and Hidden Dimensions for Batch Size 2048**. Each sub-figure represents the performance across hidden dimensions of 256, 512, 1024, and 2048. Dashed lines highlight where FlashSampling significantly surpasses Naive Sampling in memory efficiency. The smaller the metric, the better.

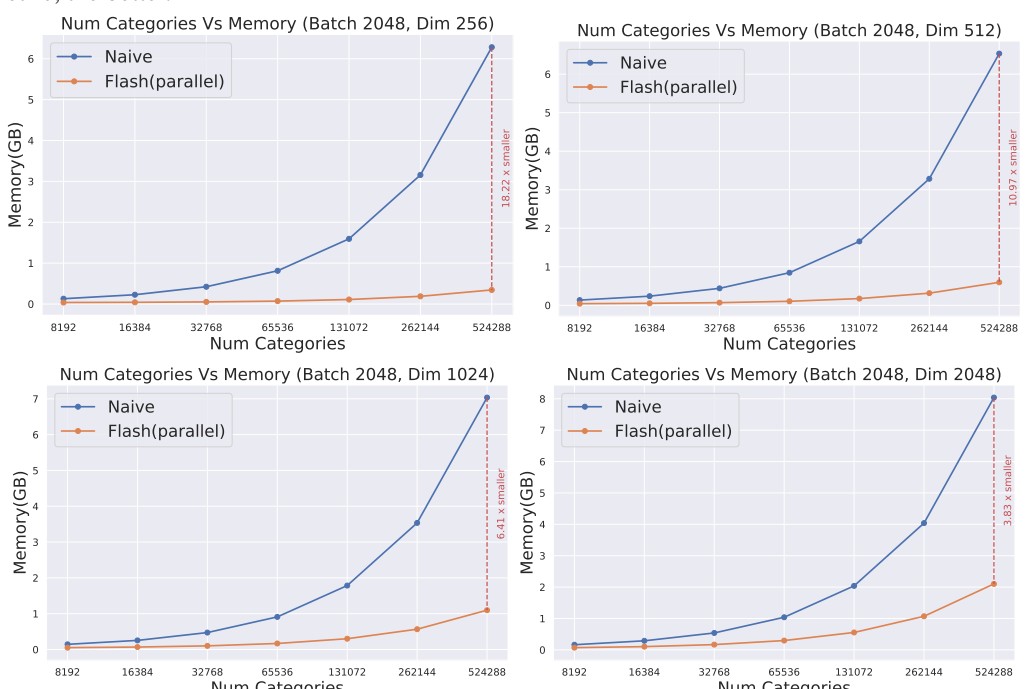

Table 2: **Memory Consumption of Naive Sampling, Parallel FlashSampling, and Sequential FlashSampling Across Vocabulary Sizes (8K-512K) in 128 and 256 Dimensions**. Dashes indicate out-of-memory errors. Memory values in the table are measured in gigabytes (GB) and must not exceed 80GB. Lower values are preferred.

| Method\Vocab. | Dim | 8,192 | 16,384 | 32,768 | 65,536 | 131,072 | 262,144 | 524,288 |
|---|---|---|---|---|---|---|---|---|
| Naive | 128 | 3.07 | 6.07 | 12.08 | 24.10 | 48.13 | - | - |
| Flash-parallel | 128 | 0.10 | 0.13 | 0.20 | 0.34 | 0.63 | 1.19 | 2.31 |
| Flash-sequential | 128 | 0.07 | 0.07 | 0.08 | 0.09 | 0.13 | 0.19 | 0.31 |
| Naive | 256 | 3.10 | 6.11 | 12.13 | 24.16 | 48.22 | - | - |
| Flash-parallel | 256 | 0.13 | 0.17 | 0.25 | 0.41 | 0.72 | 1.34 | 2.59 |
| Flash-sequential | 256 | 0.10 | 0.11 | 0.13 | 0.16 | 0.22 | 0.34 | 0.59 |

only 1/18 of the memory. As detailed in Table 1 and Table 2, FlashSampling(sequential) is slightly slower than the baseline—within 10%—yet impressively consumes only 1% of the memory. Even when scaling cagetory sizes up to 512K, FlashSampling(sequential) maintains memory usage below 1 GB. FlashSampling(parallel) consistently outperforms Naive Sampling by a significant margin. However, due to limited parallelism, FlashSampling(sequential) experiences a slowdown in calculations, which is slated for optimization in the upcoming version.

## 4.2 FAST SAMPLING: END-TO-END COMPARISON

In this section, we delve into the outcomes of implementing FlashSampling for LLM inference on the LLaMA-8B-Insturct (Dubey et al., 2024), conducted within an 8 H100 GPU environment with tensor parallel size = 8 based on gpt-fast (Liang et al., 2024).

In LLM inference, the 'LmHead' layer typically employs tensor parallelism, which splits along the vocabulary dimension Kwon et al. (2023). For example, assuming a vocabulary size of 8k and a

Figure 4: **Token per seconds (TPS) Comparison of Naive Sampling and FlashSampling(distributed) across sequence Lengths and Batch Sizes in LLaMA-8B**. Each sub-figure displays performance for batch sizes of 4, 8, 16, and 32, with higher TPS values indicating better performance.

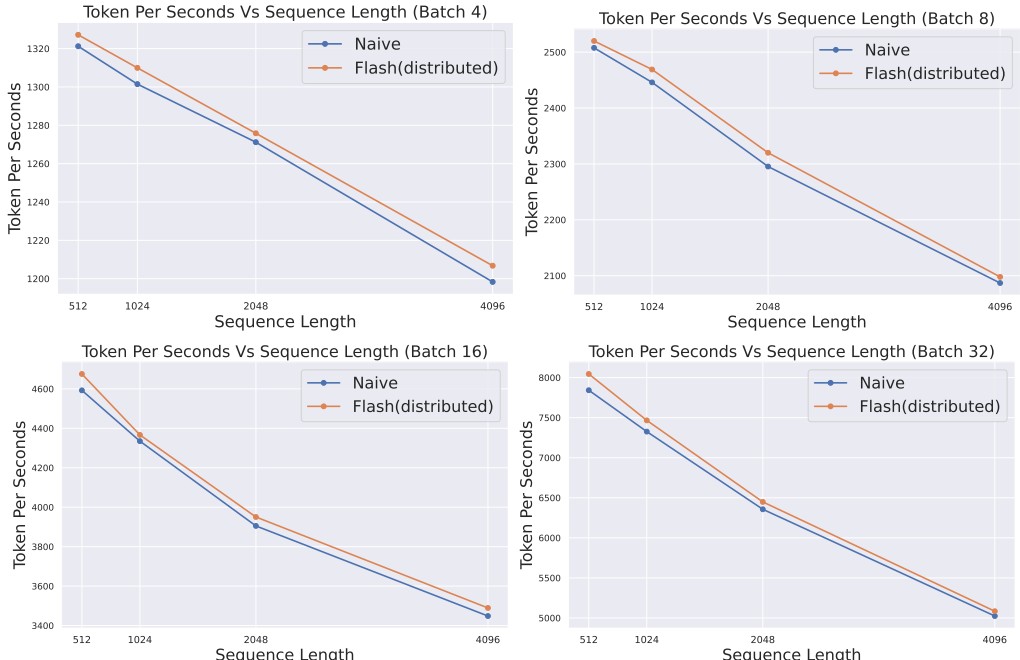

weight matrix $\mathbf{W} \in \mathbb{R}^{d \times 8000}$, with a tensor parallelism degree of 8 (and the same number of GPUs), GPU with rank $k$ will have the weight matrix slice $\mathbf{W}[:, 1000k : 1000k + 1000]$.

During sampling, each GPU (rank $k$) computes the logits for positions $1000k$ to $1000k + 1000$, followed by a 'gather' operation where GPU rank 0 gathers the complete logits and performs the sampling. Using FlashSampling (distributed), we have significantly minimized communication overhead, reducing it by a factor of 1000—or more specifically, $V$/num of tensor parallel size). The advantageous outcomes of this strategy are demonstrated in Fig. 4 and Fig. 5. It can be seen that FlashSampling (Distributed) significantly reduces memory usage while achieving faster speeds.

### 4.3 EMPIRICAL VERIFICATION IN LLM

In this experiment, we compare the generation outcomes obtained using Vanilla Sampling and FlashSampling on LLaMA3-8B-Insturct. The prompt used for generating text was ***Hello, my name is***. The results indicate that the outputs from the FlashSampling method are comparable to Vanilla Sampling, aligning with the theoretical analysis.

**Vanilla Sampling** :

<| begin_of_text |>Hello, my name is Somer. I wanted to share a little about my life . I'm a busy individual with a full schedule of work, family and personal responsibilities . I love my pets and spending time with them. I also enjoy spending time in my garden and trying to grow my own herbs and vegetables . I have a huge passion for cooking and experimenting with new recipes . One of my favorite things to do is to try out new recipes on my friends and family and see the looks on their faces when they try something new and delicious . As for travel , I'm an avid adventurer and love to explore new places . I've been to many beautiful destinations and can't wait to add more to my list . My goal in life is to live a life of purpose and to help others , whether it be through my cooking, my gardening, or just simply being a good friend and family member. I believe that life is too short to waste time and I strive to make the most of every moment.

Figure 5: **Memory consumption (measured in GB) of Naive Sampling and FlashSampling(Distributed) Across Sequence Lengths and Batch Sizes in LLaMA-8B**. Each sub-figure displays performance for batch sizes of 4, 8, 16, and 32.

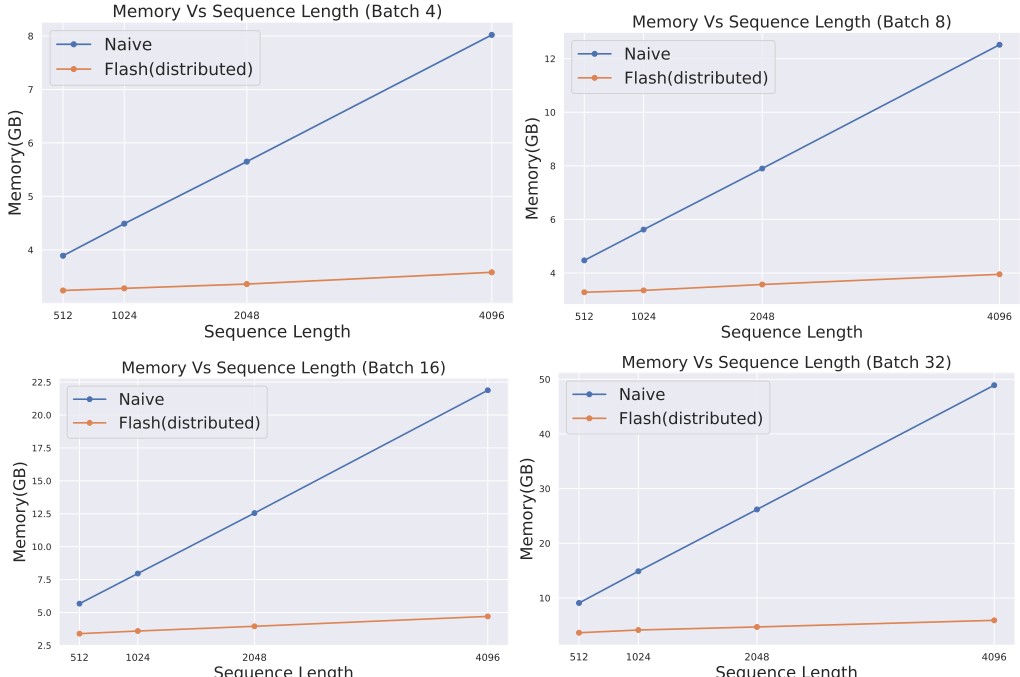

**FlashSampling** :

<| begin_of_text |>Hello, my name is Somerotos. I am an artist and designer from Mexico. I am a bitpace artist, which means that I work with a mix of traditional and digital media. I lovekeleton and skeleton art, but I also enjoy working with other themes and styles. I am a bit of a recommend artist, and I am always looking for new ways to express myself and push the boundaries of what is possible with my art. I am excited to share my art with you and hope that you will enjoy it.\nI am a bit of a creative person, and I love to experiment with different techniques and styles. I have been working as a artist for many years, and I have developed my own unique style and voice. I am a bit of a recommend artist, and I am always looking for new ways to express myself and push the boundaries of what is possible with my art.

## 5 CONCLUSION

In this paper, we have introduced FlashSampling, a novel sampling method designed to mitigate the computational and memory burdens of traditional softmax-based approaches in discrete spaces. By partitioning categories into distinct groups and leveraging the Gumbel-Max trick, FlashSampling circumvents the need for softmax computation while maintaining mathematical equivalence to conventional sampling strategies. Our method is substantiated by both mathematical proofs and empirical validations, demonstrating up to 384% faster sampling times and an impressive 1822% reduction in memory consumption. These significant enhancements underscore the potential of FlashSampling to optimize a wide array of applications—including language models, reinforcement learning, VAE, GAN, and neural architecture search—offering a more efficient alternative for future research and development.

## 6 ETHICS AND REPRODUCIBILITY STATEMENT

This research presents FlashSampling, an algorithm designed to enhance computational efficiency and reduce memory usage in machine learning sampling tasks. By minimizing resource requirements, FlashSampling can decrease energy consumption and make advanced computational techniques more accessible. While we identify no direct ethical concerns with FlashSampling, we recognize the potential for misuse and encourage users to consider the wider societal and ethical impacts of their applications.

To ensure reproducibility, we will open-source the FlashSampling algorithm along with all related code and experimental details upon publication. The publicly available datasets and detailed usage instructions are also provided to help other researchers replicate our results and apply the methodology to their own projects.

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
