# OpenReview forum: "FlashSampling: Fast and Memory-Efficient Exact Sampling with Group-Gumbel-Max"
_ICLR.cc/2025/Conference — Submitted to ICLR 2025_

### Official Review · Reviewer_R7XW · 2024-10-30

**Soundness:** 3
**Presentation:** 2
**Contribution:** 2
**Rating:** 3
**Confidence:** 3

**Summary:**

This paper proposes FlashSampling, a computationally efficient sampling framework by (1) using the Gumbel-Max trick on logit vectors to remove softmax computation and (2) parallel computing the logit vectors by decomposing the category project matrix into distinct groups. The authors show FlashSampling can reduce sampling latency and memory cost of the standard Softmax sampling while keeping a comparable sampling quality.

**Strengths:**

- I like the motivation of this paper. Deep Neural Network (DNN) often struggles with high computational requirements in the real world, the proposed method can help to reduce the latency and hardware requirement of the DNN sampling method.
- The experimental results show the proposed method has a comparable sampling quality while significantly faster and less memory consumption than the Softmax sampling when the categorical (classes) number increases.

**Weaknesses:**

- The theoretical contribution of this paper is weak. Prop.3.1 is straightforward because, from Eq.1, $z$ is defined as the highest element, so using with and without Softmax normalization is equivalent. There is no formal proof of Prop.3.1.1. The mathematical presentation in this paper is also poor for me (details are in the Miscellaneous). Last but not least, I would suggest the authors provide clear proof evidence for every Proposition, e.g., move derivation from Eq.1 to L-225 to a proof below the statement in Prop.3.1.
- The experimental results are not convincing enough. Firstly, the detailed hardware (CPUs, GPUs, etc.) and experimental settings (how to measure latency, variants between latency measurements, etc.) are not mentioned. Secondly, the reported results are not measured with multiple runs, significant tests, and across different hardware settings. Thirdly, the sampling quality evaluation in Section 4.3 is poor. It lacks the number of qualitative experiments and quantitative measurements.
- Miscellaneous: $p_k$ corresponds to the logit vector $y$ in L-216, why $p_k$ in L-240 relate to the feature embedding $x$? In L-245 (Prop.3.1) sampling what from $C(\mathbf{p})$  is equivalent to sampling $z_k$ from $C(\mathbf{p}\_k)$? In Eq.1, $\arg\max_{k=0}^{V-1}$? Eq. of $p_k$ in L-240, $\sum_{j=0}^{d-1}\exp(x_j)$ or $\sum_{j=0}^{V-1}\exp(y_j)$? Caption in Fig.2 (measured in ms/batch)? L-21 Gumbel-Max?

**Questions:**

1. What is the difference between Vanilla Sampling in Section 4.3 and Naive Sampling in Fig. 1?
2. Can we compare FlashSampling with distributed versions of the Softmax sampling by using the same group technique to avoid materializing logits?
3. How about the computational efficiency performance of FlashSampling with only the Gumbel-Max trick (i.e., without the group technique)?
4. Given there is a significant difference between the two paragraph outputs of Vanilla Sampling and FlashSampling, how can we subjectively say the "FlashSampling method are comparable to Vanilla Sampling"? Could the authors use some quantitative metrics to compare?

---

### Official Review · Reviewer_kUhw · 2024-11-01

**Soundness:** 1
**Presentation:** 1
**Contribution:** 1
**Rating:** 1
**Confidence:** 4

**Summary:**

The paper introduces FlashSampling, a method designed to bypass the computation of the softmax operation when sampling from categorical distributions. By leveraging the Gumbel-Max trick and a group-based sampling strategy, FlashSampling achieves improvements in speed and memory efficiency.

**Strengths:**

FlashSampling uses the Gumbel-max trick to avoid computing the denominator of the softmax, which saves time. FlashSampling is more scalable than softmax for very large scale experiments.

**Weaknesses:**

The paper claims that the softmax operation introduces significant computational and memory overhead, but it doesn’t provide a clear quantification of these costs relative to the rest of the sampling process. Without specific data showing softmax as the main bottleneck, the motivation for FlashSampling seem underdeveloped. This could raise questions about whether the performance gains attributed to FlashSampling are impactful.

The paper’s writing is dense with algorithms and code. Simplifying or consolidating the presentation of algorithms, and using a more selective approach to propositions and proofs, would improve readability.

**Questions:**

In the literature review, I suggest adding a discussion that directly compares this work to existing approaches. This would enhance the reader’s understanding of the motivation and contributions of the paper. Specifically, how does your method differ in terms of computational and memory efficiency from previous methods, and what new insights does it offer? Explicit comparisons with prior work would clarify both the contribution and relevance of this work

I would recommend moving Section 3.1 to a preliminaries section, or even incorporating it within the introduction. Additionally, explicitly introducing the softmax equation at this point, along with the categorical variables and class probabilities, would help introduce concepts necessary for understanding FlashSampling.

The authors mention that computing the softmax introduces significant computational and memory overheads. However, quantifying these overheads with concrete metrics would strengthen this claim. Such numbers would clarify that softmax is indeed a bottleneck. In Figure 4, it appears that the softmax computation is relatively minor compared to the overall sampling process. Could the authors clarify how much of a bottleneck softmax is in practice, relative to the preceding operations?

Proposition 3.1 could potentially be simplified, as the derivation is primarily algebraic and could be presented in a single line. I suggest presenting the Gumbel-Max trick more concisely, perhaps with a brief, one-line derivation, to improve readability. Additionally, introducing the variables $y_k$ and the categorical distribution $\mathbf{C}$ explicitly would enhance clarity. Similarly, the group sampling approach described in Proposition 3.1.1 and Algorithm 6 appears to be a straightforward result and might not require the formality of a proposition.

According to Figure 2, the time advantage of FlashSampling appears minor for categories smaller than 50,000, which is a typical range for large language models. Are there known practical applications where the number of categories would be as high as 524,288? If so, providing real-world examples where extremely large category sizes are relevant would help justify the scalability focus of FlashSampling.

---

### Official Review · Reviewer_F6pz · 2024-11-03

**Soundness:** 1
**Presentation:** 1
**Contribution:** 3
**Rating:** 3
**Confidence:** 3

**Summary:**

This paper sets out to improve the efficiency of (exact) categorical sampling. The computational efficiency is addressed using Gumbel-Max sampling (a well-known technique) and the memory efficiency by a novel two-stage group sampling strategy. The proposed algorithm, FlashSampling, is presented in both sequential and parallel settings and evaluated empirically.

**Strengths:**

The strengths of the paper are its usefulness, simple idea, and method presentation.
- The proposed method is a drop-in replacement for standard softmax sampling with better performance both in terms of computation and memory use which should be interesting to practitioners.
- The method is well-motivated and relevant proofs are present (I have not verified these in detail).
- The method is presented clearly in text, pseudocode, and Pytorch-like code.

**Weaknesses:**

The weaknesses of the paper are its related works and limited experiments with significant details missing.
- It’s difficult to understand the experiments section. In section 4.1, it is not clear what the actual experiment is (a toy example?). Furthermore, it is not clear how exactly the baseline “Naive” is implemented. This is important to address and affects the paper's overall soundness as this experiment is the basis for the empirical speedup and memory consumption claims.
- In general, the related works section covers a broad scope, but it is difficult to see the connection to this work: MCMC, SMC, differentiable sampling, and variational inference (methods that optimize the distribution's parameters) are only vaguely related to this work. At the same time, these works could warrant discussion:
    - Qi et al. [1] propose a method for faster Gumbel-Max sampling when drawing multiple samples
    - Sampled softmax [2] is a method used to circumvent the cost of large cross-entropy calculations, which is a related problem to the cost of sampling large softmax distributions.
- FlashSampling is only evaluated for one application (sampling Llama in section 4.2) despite the method’s claimed potential for broader use (line 536). In this application, it achieves significant memory savings and a small speedup. The much smaller speedup achieved for sampling Llama (fig. 4) is not discussed. Is this due to the sampling making up a small part of the generation process? What does this imply for other applications? A discussion of the method's limitations might answer some of these questions.

**References**

[1] Y. Qi, P. Wang, Y. Zhang, J. Zhao, G. Tian, and X. Guan, “Fast Generating A Large Number of Gumbel-Max Variables”, WWW 2020

[2] Ankit Singh Rawat, Jiecao Chen, Felix Yu, Ananda Theertha Suresh, and Sanjiv Kumar, “Sampled Softmax with Random Fourier Features”, NeurIPS 2019

**Questions:**

- Line 047: “...continuous space sampling in diffusion models”. The citations following this statement are not about sampling in diffusion models, and Tucker et al. (2017) is about differentiable discrete sampling. What is meant by this sentence and citations?
- When discussing speedups, the claim “up to X% faster” (best case) is often less interesting than “X% faster on average” (average case). Is it possible to include repeated experiments to compute relevant statistics of the speedup? The same goes for memory consumption.
- Consider adjusting the name “FlashSampling” slightly. The algorithm's scope is not sampling in general, it is specifically sampling softmax-parameterized categorical variables.
- Why are the example outputs of Llama in the main text of the paper? If the proposed method is provably equivalent to standard sampling, there is no need to compare the quality of outputs. Regardless, comparing the quality of outputs would require some quantitative metrics.
- Is the naive sampling in the benchmarks distributed like in the dist_sample pseudocode or sequential?

**Minor suggestions that do not individually affect the score**
- Line 020: “Gumble -> Gumbel”.
- Line 097: “Sampling in Contiguous Space” -> “Sampling in Continuous Space”.
- Line 285: “Put every thing together” -> “Putting everything together”.
- Line 293: This sentence seems incomplete and somewhat out of place. I think it can just be removed.
- Line 301 and 314: The Pytorch-like pseudocode should have some title that can be referred to, “Listing 1” for instance.
- Line 368: “Experitments” -> “Experiments”.
- Line 374: Remove “vividly”.
- Line 469: “LLaMA3-8B-Insturct” -> “Llama 3 8B Instruct”.
- Line 570: “COURNET” -> “Cournet”.
- First paragraph of the introduction: multiple examples are brought up to highlight the importance of discrete sampling. However, policies in reinforcement learning, VAEs, and GANs most often use continuous distributions. VAEs and GANs are introduced with references to VQ-VAE and SeqGAN instead of their usual meaning which should to be clarified.
- Many of the in-text citations should be in parentheses. E.g. Line 035: “Sutskever (2014)” -> “(Sutskever, 2014)”.
- Section 3.2.1 starts with “Using” and 3.2.2 with “Use”. Stick to one of the two.
- Algorithm 5 is difficult to read due to the spacing and line breaks.

---

### Official Review · Reviewer_4kUa · 2024-11-05

**Soundness:** 2
**Presentation:** 2
**Contribution:** 2
**Rating:** 3
**Confidence:** 3

**Summary:**

This paper studies the question of efficient sampling probabilities, an operation that is frequently done in deep learning (such as in the prediction of next-token probabilities).  The authors proposes an algorithm (FlashSampling) that leverages the group-Gumbel-group max.  The authors then compare this with the naive softmax algorithm and find memory and sampling time improvements.

**Strengths:**

The paper studies the question of sampling, which clearly has applications in, e.g., the video domain.  Efficient sampling can be critical to obtaining good performance on some algorithms, so the overall research questions the authors investigate is an important one.

**Weaknesses:**

I am not convinced by the efficacy of this method; in particular, it is not clear that the relevant baseline is a naive softmax algorithm.  More extensive experiments are needed to establish the efficacy of the method; for example, some relevant baselines would be e.g., parallelized softmax (as done in FlashAttention) or against MCMC methods or common speed-up methods in LLM inference, such as speculative decoding.  The paper would also benefit for a more rigorous time and space complexity analysis, comparing to other SoTA sampling methods; an application to the video domain where sampling is useful in e.g., compositional tasks, would also be necessary for me to consider acceptance.

The paper can also benefit with some major revisions for readability and spelling (e.g., section names are misspelled such as 'Experitments' in section 4).

**Questions:**

(See above)

---

### Meta-Review · Area_Chair_jSgD · 2024-12-19

**Metareview:**

This paper proposes an approach to exact categorical sampling from the softmax, using the Gumbel-max trick with a grouping strategy.  While the reviewers found the paper well motivated and very relevant to the community, they unfortunately all voted to reject the paper. Common criticisms across reviews included:

- Experimental rigor:  Concerns about the limited experiments, lack of comparison to proper baselines, insufficient details on hardware/settings, and the lack of quantitative evaluations of sampling quality.

- Lack of clarity and writing: Issues with understanding experiments, code, and mathematical explanations, plus the need for clearer proofs and presentation of the proposed approach.  The writing style is dense, which makes the paper harder to understand.

- Motivation/Novelty: Questions about how the claimed softmax overhead is calculated/quantified; there is a demand for proper justification for the focus on the softmax.

The authors did not provide a response during the discussion period.  It seems that the paper is not quite ready for publication.  Therefore the recommendation is to reject the paper.  Hopefully the reviews will be helpful to strengthen the paper for a future submission.

**Additional Comments On Reviewer Discussion:**

The authors did not provide a response during the discussion period.

---

### Decision · Program_Chairs · 2025-01-22

Reject